# Proteomic profiling of serum extracellular vesicles identifies diagnostic markers for echinococcosis

Xiaola Guo [1☯*], Shuai Wang[1☯], Junmei Zhang[1], Rui Li[1], Yong'e Zhang[1], Zhengrong Wang[2], Qingming Kong[3], William C. Cho [4], Xianghong Ju[5], Yujuan Shen[6], Lingqiang Zhang[7], Haining Fan[7], Jianping Cao[6*], Yadong Zheng[8*]

**1** State Key Laboratory of Veterinary Etiological Biology, Key Laboratory of Veterinary Parasitology of Gansu Province, Lanzhou Veterinary Research Institute, Chinese Academy of Agricultural Sciences, Lanzhou, China, **2** State Key Laboratory for Sheep Genetic Improvement and Healthy Production, Xinjiang Academy of Agricultural and Reclamation Science, Shihezi, China, **3** Institute of Parasitic Diseases, School of Biological Engineering, Hangzhou Medical College, Hangzhou, China, **4** Department of Clinical Oncology, Queen Elizabeth Hospital, Hong Kong SAR, China, **5** Department of Veterinary Medicine, College of Agriculture, Guangdong Ocean University, Zhanjiang, China, **6** National Institute of Parasitic Diseases, Chinese Center for Disease Control and Prevention, (Chinese Center for Tropical Diseases Research), Key Laboratory of Parasite and Vector Biology, National Health Commission of the People's Republic of China; Shanghai, China, **7** Department of Hepatopancreatobiliary Surgery, Qinghai University Affiliated Hospital, Qinghai Province Key Laboratory of Hydatid Disease Research, Xining, China, **8** Key Laboratory of Applied Technology on Green-Eco-Healthy Animal Husbandry of Zhejiang Province, Zhejiang Provincial Engineering Laboratory for Animal Health Inspection & Internet Technology, Zhejiang International Science and Technology Cooperation Base for Veterinary Medicine and Health Management, China-Australia Joint Laboratory for Animal Health Big Data Analytics, College of Animal Science and Technology & College of Veterinary Medicine of Zhejiang A&F University, Hangzhou, China

☯ These authors contributed equally to this work.

\* guoxiaola@caas.cn (XG); caojp@chinacdc.cn (JC); zhengyadong@zafu.edu.cn (YZ)

**Data Availability Statement:** All relevant data are within the paper and its Supporting Information files.

## Abstract

Echinococcosis is a parasitic disease caused by the metacestodes of *Echinococcus* spp. The disease has a long latent period and is largely underdiagnosed, partially because of the lack of effective early diagnostic approaches. Using liquid chromatography-mass spectrometry, we profiled the serum-derived extracellular vesicles (EVs) of *E. multilocularis*-infected mice and identified three parasite-origin proteins, thioredoxin peroxidase 1 (TPx-1), transitional endoplasmic reticulum ATPase (TER ATPase), and 14-3-3, being continuously released by the parasites into the sera during the infection via EVs. Using ELISA, both TPx-1 and TER ATPase were shown to have a good performance in diagnosis of experimental murine echinococcosis as early as 10 days post infection and of human echinococcosis compared with that of control. Moreover, TER ATPase and TPx-1 were further demonstrated to be suitable for evaluation of the prognosis of patients with treatment. The present study discovers the potential of TER ATPase and TPx-1 as promising diagnostic candidates for echinococcosis.

**Funding:** The study was financially supported by grants from the State Key Laboratory of Sheep Genetic Improvement and Healthy Production (Grant No. MYSKLKF202003 to XG), the Agricultural Science and Technology Innovation Program (Grant No. CAAS-ASTIP-2016-LVRI-03 to SW), and Scientific Research and Development Talent Fund of Zhejiang Agriculture and Forestry University (Grant No. 2021LFR038 to YZ). The funders had no role in study design, data collection and analysis, decision to publish, or preparation of the manuscript.

**Competing interests:** The authors have declared that no competing interests exist

## Author summary

Echinococcosis, also known as hydatid disease, is one of the neglected zoonotic diseases. *Echinococcus multilocularis* and *Echinococcus granulosus sensu latu* are the causative agents responsible for alveolar echinococcosis and cystic echinococcosis, respectively. Alveolar echinococcosis is mainly endemic in the areas of the northern hemisphere, whereas cystic echinococcosis has a worldwide distribution. The disease has a long latent period of up to 5 to 10 years and is largely underdiagnosed, partially due to the lack of effective early diagnostic approaches. In recent years, an emerging role of EVs in intercellular communication and diagnosis has been recognized. Here, we employed a proteomic approach to identify parasite-derived proteins in the serum EVs from *E. multilocularis*-infected mice and assessed their diagnostic values. This study signify the role of EVs for the identification of diagnostic candidates by the discovery of two identified proteins, TER ATPas and TPX-1. First results indicate their diagnostic and prognostic values in experimental murine and human echinococcosis.

## Introduction

Echinococcosis, also known as hydatid disease, is one of the neglected zoonotic diseases recognized by World Health Organization (WHO) [1]. *Echinococcus multilocularis* and *Echinococcus granulosus sensu lato* are the causative agents responsible for alveolar echinococcosis (AE) and cystic echinococcosis (CE), respectively. Wild rodents and sheep generally act as intermediate hosts for *E. multilocularis* and *E. granulosus*, respectively. Humans are accidental intermediate hosts and are infected by ingestion of food or water contaminated with eggs. AE is mainly endemic in the areas of the northern hemisphere, whereas CE has a worldwide distribution [2]. Both AE and CE are highly endemic in western China, with estimated 157,500 patients and 50 million people at risk [3]. The disease has a long latent period of up to 5 to 10 years and is largely underdiagnosed, partially due to the lack of effective early diagnostic approaches.

Extracellular vesicles (EVs) are secreted by most cell types and exist in almost all-biological fluids, including serum/plasma and urine [4,5]. EVs were originally identified in reticulocytes as a supposed garbage-disposal mechanism [6]. Currently, EVs have been recognized for their remarkable role in intercellular communication [7–9], immune responses [10,11], and diagnosis of infectious diseases [12,13]. EVs have been characterized in a number of helminth parasites and these EVs can be efficiently internalized by recipient cells, thus manipulating the host immune responses and contributing to pathogenesis [14–17]. The characterization of parasite-derived EVs in the fluids of infected hosts might represent an efficient approach to identify new candidates for echinococcosis diagnosis.

In the present study, we employed a proteomic approach to identify parasite-derived proteins in the serum EVs from *E. multilocularis*-infected mice and assessed their diagnostic values. The study paves the way for the use of EVs to identify of diagnostic candidates and potentiates the applications of TER ATPase and TPx-1 in the early diagnosis and prognostic evaluation of echinococcosis.

## Materials and methods

### Ethics statement

Animal studies have been approved by the Animal Ethics Committee of Lanzhou Veterinary Research Institute, Chinese Academy of Agricultural Sciences. All animals used for the

experiments were raised and handled in strict accordance with the Good Animal Practice Requirements of the Animal Ethics Procedures and Guidelines of the People's Republic of China. Human studies were approved by the Research and Ethics Committee of Qinghai University Medical College (QHDX-2016–03). Written informed consent was obtained from all adult research participants as well as from parents/guardians of all children aged between 7 and 18 years.

## Parasite infection and sample collection

Mice were injected at two different time points intraperitoneally with *E. multilocularis* protoscoleces as described previously [18]. A first experimental infection of mice with *E. multilocularis* was established for EV isolation. Six to eight-week-old male BALB/c mice were divided randomly into two groups. One group (n = 60) was intraperitoneally injected with 1,000 *E. multilocularis* protoscoleces and the other group (n = 60) was inoculated with phosphate-buffered saline (PBS) solution as a control. The mice were euthanized at 30- (n = 20), 60- (n = 20), and 90-days (n = 20) post infection. The uninfected mice at each sampling time point (n = 20) were used as controls. The sera of 10 mice of each group were pooled, giving rise to three of EV-infected mice samples and three of EV-uninfected mice samples for each time point. A second experimental infection of mice with *E. multilocularis* was designed for assessment of the enzyme-linked immunosorbent assay (ELISA) established in the study. Male BALB/c mice (n = 64) were injected with *E. multilocularis* protoscoleces as above and 50 μL of blood samples were collected by tail vein bleeding at 0-, 5-, 6-, 7-, 8-, 9-,10-, 15-, 20-, and 30-days post infection, and added into the microcentrigue tubes containing 50 μL physiological saline. The pre-infection sera (at 0-days post infection) were used as control. Samples were centrifuged at 2,000×g for 10 min at 4˚C. Serum samples were immediately transferred into a clean tube and frozen at −80˚C for late use.

## Human serum samples

The patients with echinococcosis were confirmed by postoperative pathology, clinical ultrasound, and enzyme-linked immunosorbent assay (ELISA)/indirect hemagglutination assay (IHA) based on previous literature [19]. A total of 94 serum samples of echinococcosis patients, 59 cured patients who had taken albendazole and fully recovered confirmed by ultrasound and Em18 protein-based ELISA [20], and 65 serum samples from healthy volunteers were collected. Except for one patien who had cyst(s) in the spleen, all cases had cyst(s) (> 5 cm in diameter) in the liver. Whole blood samples were collected for serum isolation. After centrifugation at 2,000×g for 10 min at 4˚C, serum samples were immediately transferred into a clean tube and frozen at −80˚C until further use.

## Enrichment and characterization of EVs

Serum EVs were isolated as previously reported with some modifications [21,22]. In brief, sera were first centrifuged at 12,000 ×g for 60 min at 4˚C, filtered using 0.22 μm filters and then centrifuged at 100,000 ×*g* for 2 h. Then the EVs-containing pellets were washed using filtered PBS, centrifuged at 100,000 ×*g* for 2 h, and finally resuspended in filtered PBS for further analysis.

The size distribution of the enriched EVs was determined by nanoparticle tracking analysis (NTA) with a Zetasizer Nano (Malvern, Instruments, Malvern, UK) as described previously [15]. The morphology of the EVs was characterized by transmission electron microscopy (TEM) as described previously [15]. Briefly, 10 μL of the enriched EVs were diluted at 1:5 in filtered PBS and coated onto a carbon-coated copper grid. After incubation for 5 min at room

temperature (RT), the grids were negatively stained with 3% phosphotungstic acid for 5 min at RT. The grids were then visualized under a JEM–2010 transmission electron microscop (JEOL, Tokyo, Japan).

### Immune transmission electron microscope (ITEM)

ITEM was performed based on previous literature [23]. Briefly, EVs were resuspended in PBS and transfer onto formvar-coated gold grids (200 mesh, Agar Scientific), followed by fixation in 4% paraformaldehyde for 30 min. The grids were incubated with 100 μl of 50 mM glycine and the grids were then blocked with 5% bovine serum albumin (BSA). The primary antibody (1:200 dilution) against emu-14-3-3 (previously prepared in our lab) was added and incubated for 45 min. Subsequently, the grids were washed four times in PBS buffer and gold-labeled secondary antibodies (Jackson Immunoresearch, dilution 1:20) were incubated for 30 min. Finally, the grids were negatively stained by 3% phosphotungstic acid directly. Grids were visualized under a JEM–2010 transmission electron microscop (JEOL, Tokyo, Japan).

### Protein digestion and mass spectrometry

Protein sequencing was carried out in BGI-Shenzhen (Shenzhen, China). Briefly, EVs-containing pellets were extracted using RIPA buffer with protease inhibitor Cocktail, and then quantified using the Bradford Assay. EV proteins were reduced with 10 mM dithiothreitol (DTT), and alkylated with 55 mM iodoacetamide (IAM). A total of 10 μg of proteins were digested by addition of 0.25 μg trypsin (Gibco) at 37°C for 12 h. After trypsin digestion, the resulting peptides were desalted using Strata X columns and vacuum-dried. The dried peptide samples were solubilized in mobile phase A (2% acetonitrile:0.1% formic acid). The desalted peptides were subjected to nanoLC-MS/MS on tandem self-packed C18 column (20 cm in length×7.5 μm internal diameter, pore size of 3 μm) coupled to tandem mass spectrometry in a Triple TOF 5600 (SCIEX) (SCIEX, Framingham, MA, USA). An linear gradient of mobile phase B (v/v) was used as follows: (0-5min, 5% phase B; 5–90 min, 5–25% phase B; 90–100 min, 25–35% phase B; 100–110 min, 35%- 80% phase B; 115–116 min, 80–5% phase B, 116-120min, 5% phase B). For data acquisition, the mass spectrometer parameters were set as follows: the spray voltage of 2.3 kV and spray interface temperature of 160°C. For the confirmation of identification, the mass spectrometer performed a full MS scan under the following settings: the m/z range was 350~1,250 *m/z* with resolving power of 600,000, repeat count of 2, repeat duration of 30 s, and an exclusion duration of 90 s. The mass spectrometry data have been deposited in ProteomeXchange Consortium with an data identifier PXD036031.

### Protein identification

For protein identification, tandem mass spectra were extracted from each sample and analyzed using Mascot search engine (Mascot v2.3) by searching against the following databases: mouse protein database (88,027 sequences) obtained from Uniprot (Release 2022/02) and *E. multilocularis* genome and protein databases PRJEB122 (10,971 sequences, retrieved from WormBase ParaSite). Peptides were filtered with the following parameters: minimum peptide length of 7, a maximum false discovery rate (FDR) for peptides and proteins of 1%, a minimum score for modified peptides of 40, and a main search error of 4 ppm. Carbamidomethyl of cysteine was used as a fixed modification, and methionine oxidation was used as a variable modification. Intensity-based absolute quantification in MaxQuant (version 1.6.2.3) was performed with the identified peptides to quantify the protein abundance. Differentially abundant proteins between the two groups were compared using Student's *t*-test with a cut-off value of $p < 0.05$

and fold change > 1.5. A heatmap was then generated in the R package gplots (v1.0) using heatmap.2.

## Preparation of recombinant proteins

The gene sequences encoding thioredoxin peroxidase 1 (TPx-1) and transitional endoplasmic reticulum ATPase (TER ATPase) were retrieved from WormBase Parasite (https://parasite.wormbase.org). Using specific primers (S1 Table), the full-length coding sequences of both genes and the N-terminal fragment of *TER ATPase* gene (1–300 aa) were amplified and subsequently sequenced. PCR amplified fragments were subcloned into vector pET-28a (+) and positive plasmids were transformed into *Escherichia coli* (BL21) cells. The recombinant His tagged proteins were induced with 0.5 M isopropyl β-d-1-thiogalactopyranoside (IPTG) and purified by Ni Sepharose 6 Fast Flow (GE Healthcare, Chicago, IL, USA). The recombinant proteins were assessed using a 10% SDS–PAGE gel and quantified using bicinchoninic acid (BCA) protein assay kit (Beyotime, Jiangsu, China).

## Generation and purification of polyclonal antibodies

Each New Zealand white rabbit (n = 3) was subcutaneously immunized with 100 μg of each purified recombinant protein emulsified with and without Freund's adjuvant as described previously [24]. Immunoglobulin G was purified by protein G affinity chromatography (GE Healthcare), and the concentration was determined using BCA protein assay kit (Beyotime).

## Fluorescent immunohistochemistry and Western blotting

Liver tissue sections with hydatid cysts from *E. multilocularis*-injected mice were deparaffinized in xylene, rehydrated in graded alcohols, and soaked in 3% hydrogen peroxide, followed by antigen retrieval. After they were blocked with 5% BSA in TBS-T, the sections were incubated with anti-TER ATPase (prepared in the study) or anti-TPx-1 (prepared in the study). After washes, the sections were incubated with the Alexa Fluor 594-conjugated goat anti-rabbit secondary antibody (1:1000, Ther-moFisher Scientific, USA) for 1h at 37˚C, followed by addition of mounting medium containing a nucleus stain DAPI (Cell Signaling, USA). Images were recorded under fluorescent microscopy (Leica, USA).

For Western blotting, protein samples were separated using 10% SDS–PAGE, and then transferred onto polyvinylidene fluoride (PVDF) membranes. After blocking with 5% non-fat milk, the membranes were incubated overnight at 4˚C with 1:1,000 dilutions of anti-CD63 (Abcam, Cambridge, MA, USA), anti-TGF-β1 (AF1027, Affinity Biosciences, Cincinnati, OH, US), anti-GPATC8 (DF9522, Affinity), anti-LRP1 (DF2935, Affinity), anti-anti-MYCBP (DF10076, Affinity), anti-FBPA (previously prepared in our lab), anti-TPx-1 (prepared in the study), or anti-TER ATPase (prepared in the study). Subsequently, the membranes were washed and incubated with goat anti-rabbit IgG secondary antibody (1:10,000, KPL), and were visualized by exposing X-ray films using Pierce ECL Western Blotting Substrate (Thermo-Fisher Scientific).

## Enzyme-linked immunosorbent assay (ELISA)

Each well of 96-well plates (Corning, NY, USA) was coated with 100 μL of 50 mM sodium carbonate buffer (pH 9.6) containing 1 μg of crude antigens of *E. multilocularis* previously prepared in our laboratory [18], TPx-1, or TER ATPase at 4˚C overnight, followed by addition of 300 μL of casein blocking buffer (B6429, Sigma, St. Louis, MO, USA) for 1 h at 37˚C. After three washes in PBS with Tween-20 (PBS-T), the plates were incubated with 100 μL of mouse

sera (1:200) or human sera (1:400) for 1 h at 37˚C. The plates were subsequently washed three times in PBS-T, and incubated with HRP-conjugated goat anti-mouse IgG (1:8,000, KPL) or HRP conjugated goat anti-human IgG (1:8,000, KPL) for 1 h at 37˚C. After washing, the reactions were developed by incubation with 100 μL of TMB One Solution (G7431, Promega, Madison, WI, USA) for 15 min in the dark. The reactions were stopped by adding 50 μL of 2 M $H_2SO_4$, and the optical density (OD) was measured at 450 nm using a Spectra Max ABS Plus microplate reader (Molecular Devices, San Jose, CA, USA). For each antigen, the positive cut-off values were calculated by the OD value of each test serum (S) divided by the value OD of control sera (N). Samples with an S/N value more than 2 were considered to be positive.

For each antigen, receiver operating characteristic (ROC) curves were generated and the area under the ROC curves (AUC) was calculated. The cut-off value for each antigen was determined by the maximum Youden index of the ROC curve and calculated using Medcalc.

## Statistical analysis

The log-rank test was performed to determine the statistical significance among three groups at 30-, 60-, and 90-days post infection. The Wilcoxon rank-sum test was used for comparisons between cured patients and untreated-patients with echinococcosis. A significant difference was considered if the $p$ value was less than 0.05.

## Results

### Isolation and characterization of serum-derived EVs from *E. multilocularis*-infected mice

To understand the dynamic changes of protein components in mouse serum-derived EVs in response to *E. multilocularis* infection, we isolated and comparaed the samples at 30-, 60- and 90-day time points (30 d, 60 d, and 90 d infected and uninfected mice). Transmission electron microscope (TEM) of the EVs isolated from infected and uninfected mice was shown to be physically homogeneous and was about 60 to 110 nm in diameter (Fig 1A and 1B). Similarly, the NTA results also revealed that these EVs were highly homogeneous in size, with a diameter of 60–200 nm and a peak at approximately 110 nm (Fig 1C). There were no significant differences in the size and morphology between these EVs. It was further shown that a mammalian EV marker protein (CD63) was more abundant in these EVs (Fig 1D).

### Mouse protein components in serum-derived EVs

A total of 526 mouse proteins were identified from 6 serum-derived exosome samples (3 from infected mice and 3 from uninfected mice, S2 Table), 121 of which were commonly shared. In total, only 18 proteins (14 upregulated and 4 downregulated) were statistically differentially abundant in the EVs derived from infected mice compared with those from uninfected mice (Fig 1E and Table 1). Among them, the levels of transforming growth factor beta-1 (TGF-β1), G-Patch domain containing 8 (GPATC8), and prolow-density lipoprotein receptor-related protein 1 (LRP1) were significantly upregulated, whereas Alpha-amylase 1 (AMY1) was significantly downregulated in *E. multilocularis*-infected serum EVs compared with these in the control.

To confirm these findings, the abundance of these four proteins was validated by Western blotting. The results showed that the levels of these EV proteins were regulated in response to *E. multilocularis* infection. With the extension of infection, the levels of GPATC8 and TGF-β1 exhibited a gradual increase, whereas the level of LRP1 and AMY1 showed a gradual decrease (Fig 1F).

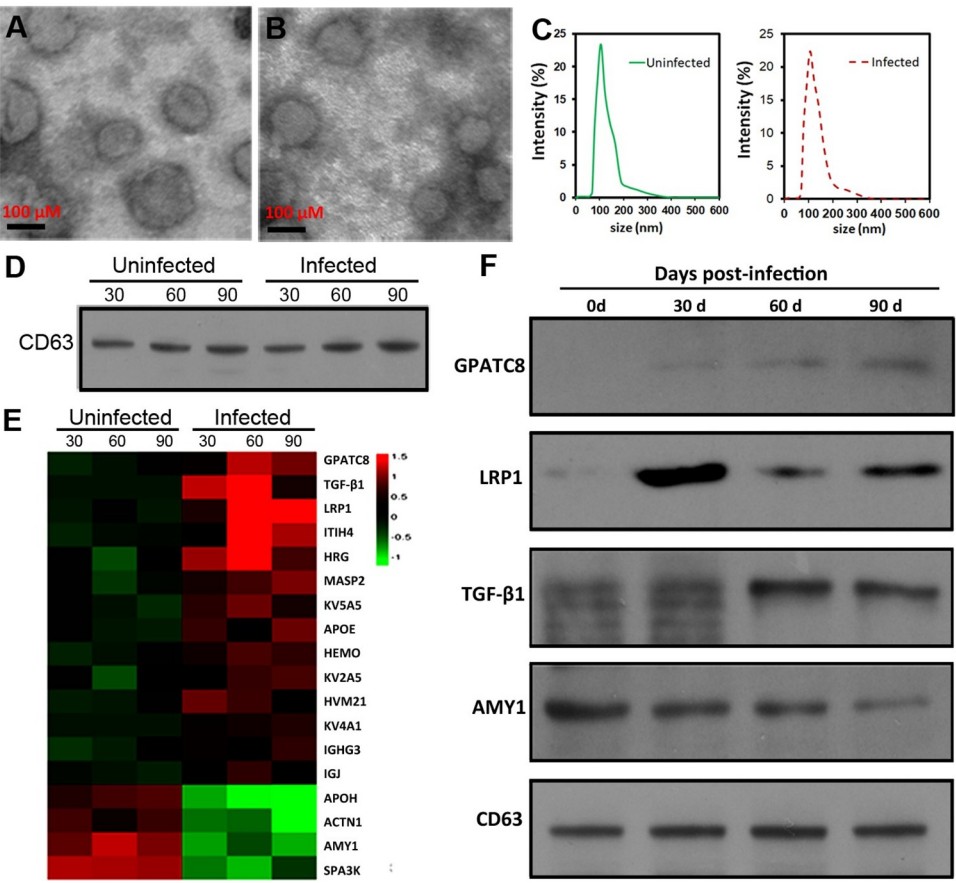

**Fig 1. The dynamic changes of mouse proteins in serum-derived exosomes in response to *E. multilocularis* infection.** (A) and (B) Transmission electron microscopy of the enriched extracellular vesicles (EVs) isolated from the sera of *E. multilocularis*-infected (A) and uninfected mice (B). (C) The diameter distribution of the purified EVs. (D) Western blotting analysis of mammalian EV-marker CD63 in the purified EVs. (E) Heatmap of differentially abundance proteins between the serum exosomes from *E. multilocularis*-infected and uninfected mice. (F) *Western blotting* analysis the abundance of TGF-β1, GPATC8, LRP1, and AMY1 during *E. multilocularis* infection.

## Parasite-origin protein components in serum-derived EVs

To determine whether *E. multilocularis*-derived EVs are present in the sera of infected mice, we performed immune electron microscopy using anti-Em14-3-3 antibodies [15]. The results showed that there was a subpopulation of parasite-derived EVs (Fig 2A). Consistently, ten parasite-origin proteins were identified, such as TPx-1, TER ATPase, fructose-1, 6-bisphosphate aldolase (FBPA), glyceraldehyde-3-phosphate dehydrogenase (GAPDH), and 14-3-3 (Table 2). Interestingly, we found that three proteins TPx-1, TER ATPase, and 14-3-3 were continuously present at all infection time points, while the rest were only detected at specific time point(s) (Table 2).

To further confirm these findings, four proteins TPx-1, TER ATPase, FBPA, and 14-3-3 were selected for validation by Western blotting. First, we used the purified recombinant TPx-1 and TER ATPase (truncated) to raise specific polyclonal antibodies, and then demonstrated that they could recognize the native *E. multilocularis* proteins (S1 Fig). Consistent with the proteomic data, Western blotting analysis showed that TPx-1,TER ATPase, and 14-3-3 were detected in the serum EVs at all infection time points, while FBPA was only detected in the

**Table 1. Summary of the differentially abundant proteins in serum-derived exosomes from *E. multilocularis*-infected mice.**

| Protein name | Protein ID | Fold change | *P* value |
|---|---|---|---|
| **Upregulated** | | | |
| Transforming growth factor beta-1 | P04202 | 16.4239 | 0.0001 |
| G-Patch Domain Containing 8 | A2A6A1 | 16.3072 | 0.0002 |
| Prolow-density lipoprotein receptor-related protein 1 | Q91ZX7 | 14.7885 | 0.0233 |
| Histidine-rich glycoprotein | Q9ESB3 | 9.9466 | 0.0041 |
| Inter alpha-trypsin inhibitor, heavy chain 4 | A6X935 | 6.1029 | 0.0142 |
| Mannan-binding lectin serine protease 2 | Q91WP0 | 4.1713 | 0.0198 |
| Ig kappa chain V-V region T1 | P01637 | 3.5643 | 0.0245 |
| Ig heavy chain V region M511 | P01790 | 3.5098 | 0.0049 |
| Apolipoprotein E | P08226 | 3.4387 | 0.0237 |
| Hemopexin | Q91X72 | 3.1622 | 0.0636 |
| Ig kappa chain V-II region 17S29.1 | P03976 | 2.9148 | 0.0001 |
| Ig kappa chain V-V region L7 | P01680 | 2.3316 | 0.0108 |
| Ig gamma-3 chain C region | P03987 | 2.3013 | 0.006 |
| Immunoglobulin J chain | P01592 | 2.1040 | 0.0239 |
| **Downregulated** | | | |
| Beta-2-glycoprotein 1 | Q01339 | 0.1426 | 0.0078 |
| Serine protease inhibitor A3K | P07759 | 0.4115 | 0.0328 |
| Alpha-amylase 1 | P00687 | 0.3502 | 0.0116 |
| Alpha-actinin-1 | Q7TPR4 | 0.3006 | 0.0125 |

EVs 90 d post infection (Fig 2B). Immunofluorescence localization showed that both TER ATPase and TPx-1 proteins were visible in the liver areas close to parasitic lesions (Fig 2C).

## Values of TPx-1 and TER ATPase in the early diagnosis of experimental murine AE

Considering their continuous presence in serum EVs, we speculated that both TPx-1 and TER ATPase could act as early diagnostic markers. To test this idea, we first checked the immuno-genicity of both TER ATPase and TPx-1 with sera from AE mice. The results showed that both TPx-1 and TER ATPase were specifically recognized by the sera at all infection time points (Fig 2D). Next, we assessed the early diagnostic values of both antigens using ELISA. The results confirmed that both TER ATPase and TPx-1 were able to detect AE mice as early as 10 days post infection, with specificity of 100% (95% confidence interval (CI): 94.4–100.0%) and sensitivity of > 96% (95% CI: 89.2–100.0%), which were significantly higher than those of the crude antigens (Fig 3 and Tables 3 and S3). Both the antigens still showed a better performance than the crude antigens in the diagnosis of AE mice 15 days post infection (Fig 3 and Table 3), while they showed no difference from crude antigens for AE mice 20 days post infection and later (Fig 3). Moreover, TER ATPase and TPx-1 had similar sensitivity and specificity in the early diagnosis of AE (Table 3).

## Values of TPx-1 and TER ATPase in diagnosis of human echinococcosis

To further assess their diagnostic values, we tested the performance of TER ATPase and TPx-1 in the diagnosis of human echinococcosis. The results showed that, compared with crude anti-gens, both TER ATPase and TPx-1 showed a better performance in the diagnosis of human echinococcosis with an AUC of 0.976 (95% CI: 0.960–0.993) and 0.936 (95% CI: 0.898–0.963) ($p < 0.01$), respectively. Moreover, the sensitivity and specificity of TER ATPase were 71.11%

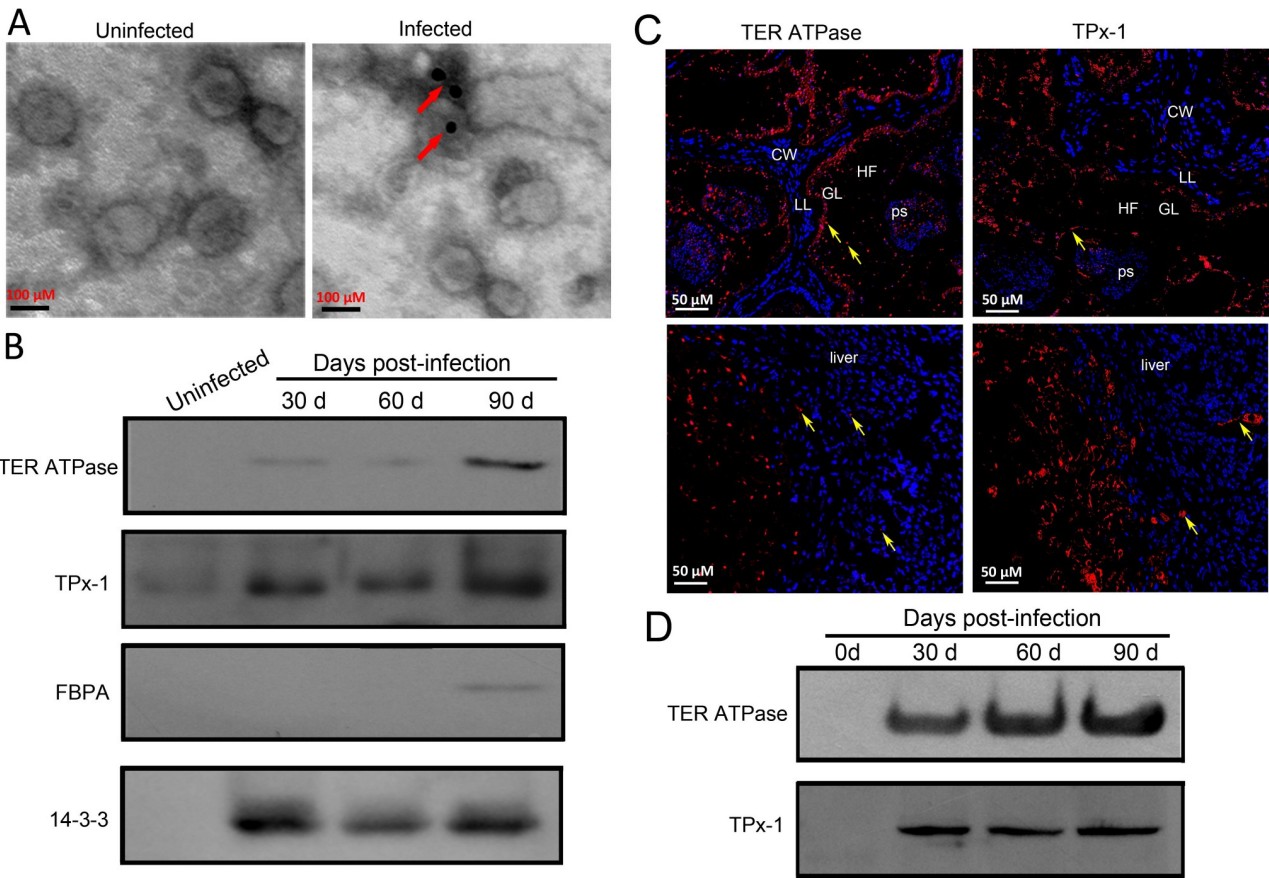

**Fig 2. Parasite-origin protein components in serum-derived exosomes in response to *E. multilocularis* infection.** (A) Immune electron microscopy of EVs isolated from the sera of *E. multilocularis*-infected and uninfected mice using anti-Em14-3-3 antibodies. The red arrows indicated the parasite-derived EVs. (B) *Western blotting* analysis of TPx-1, TER ATPase, and FBPA in the serum-derived exosomes *at 30-, 60-, 90-day post infection*. The uninfected mouse sera were used as control. (C) Immunofluorescence localization of TER ATPase and TPx-1 in the *E. multilocularis-infected mouse* liver. CW, cyst wall; GL, germinal layer; HF, hydatid fluid; LL, laminated layer; ps, protoscolexs; liver, mouse liver. The nucleus was stained by DAPI. Localization of TER ATPase or TPx-1 was indicated by yellow arrows. (D) *The recombinant TPx-1 and TER ATPase could recognize by E. multilocularis-positive sera at 30-, 60, 90-day post infection*. The pre-infection sera (at 0-day post infection) were used as control.

**Table 2. Summary of parasite proteins in serum-derived exosomes from *E. multilocularis*-infected mice.**

| Protein name | Gene accession number | Uninfected control | Days post infection | | |
|---|---|---|---|---|---|
| | | | 30 | 60 | 90 |
| Thioredoxin peroxidase | EmuJ_000791700 | - | * | * | * |
| Transitional endoplasmic reticulum ATPase | EmuJ_000471600 | - | * | * | * |
| 14-3-3 | EmuJ_000789700 | - | * | * | * |
| Fructose-bisphosphate aldolase | EmuJ_000905600 | - | - | - | * |
| Glyceraldehyde-3-phosphate dehydrogenase | EmuJ_001101100 | - | * | * | - |
| Replication factor C subunit 4 | EmuJ_000115900 | - | * | - | - |
| Aspartate aminotransferase | EmuJ_000778900 | - | * | - | - |
| Eukaryotic translation initiation factor 4 gamma | EmuJ_000391200 | - | - | * | - |
| Coronin | EmuJ_000294400 | - | * | - | - |
| Pre mRNA splicing factor CWC22 | EmuJ_000529100 | - | * | - | - |

Note: '*', detected; '-', not detected.

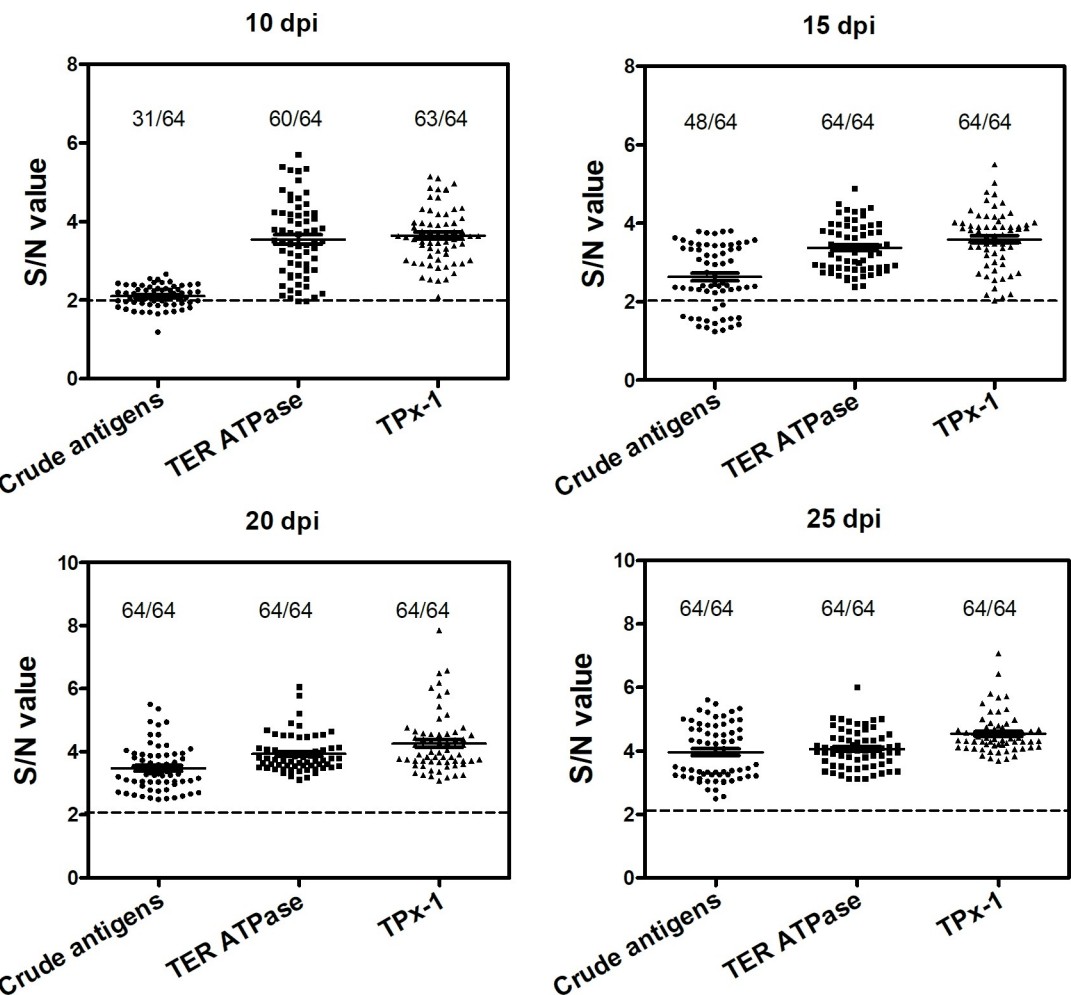

**Fig 3. The early diagnostic values of TPx-1 or TER ATPase in *E. multilocularis*-infected mice using ELISA.** For each antigen, the positive cut-off values were calculated by the OD value of each test serum (S) divided by the mean OD value of healthy sera (N). Samples with an S/N value not less than 2 were considered to be positive.

(95% CI: 61.39–80.64%) and 95.65% (95% CI: 89.01–98.78%), respectively, significantly higher than 66.67% (95% CI: 67.8–85.9%) and 94.44% (95% CI: 87.51–98.17%) of TPx-1 ($p = 0.002$, Table 4).

Taking account of the dynamic secretion of EVs, we hypothesized that the levels of antibodies against TER ATPase and TPx-1 directly reflected prognosis and so we evaluated their prognostic values using the sera of the cured echinococcosis patients. Of 59 serum samples of cured patients, only 18.64% (11/59) and 16.95% (10/59) were tested as positive using TER ATPase- and TPx-1-based ELISA, respectively, which were significantly higher than 8.47% (5/59) by using Em18 protein-based ELISA ($p < 0.001$, Table 4). Compared with TPx-1, TER ATPase showed no significant difference in the sensitivity of 81.11% (71.79–88.59%), specificity of 93.22% (95% CI: 83.54–98.12%), and an AUC of 0.95 (95% CI: 0.90–0.98) ($p = 1.000$, Table 4).

## Discussion

In recent years, an emerging role of EVs in intercellular communication and diagnosis has been recognized [25,26]. EVs can mediate intercellular communication between parasites,

**Table 3. The performance of TPx-1 and TER ATPase in the diagnosis of experimental murine echinococcosis.**

| | No. of samples | No. of positive samples | | |
|---|---|---|---|---|
| | | Crude antigens | TPx-1 | TER ATPase |
| 10-day post infection | | | | |
| Healthy mice | 64 | 0 | 0 | 0 |
| Infected mice | 64 | 31 | 60 | 63 |
| AUC (95% CI) | | 0.813 (0.741–0.885) | 0.993 (0.985–1.00) | 0.999 (0.993–1.00) |
| Sensitivity (95% CI) | | 51.6% (38.7–64.2%) | 96.8% (89.2–99.6%) | 98.4% (91.6–100.0%) |
| Specificity (95% CI) | | 98.4% (91.6–100.0%) | 100% (94.4–00.0%) | 100% (94.4–100.0%) |
| *p-value* | | | $p < 0.0001^a$ | $p < 0.0001^a$ |
| | | | $p = 0.4713^b$ | |
| 15-day post infection | | | | |
| Healthy mice | 64 | 0 | 0 | 0 |
| Infected mice | 64 | 48 | 64 | 64 |
| AUC (95% CI) | | 0.945 (0.890–0.977) | 1.00 (0.97–1.00) | 1.00 (0.97–1.00) |
| Sensitivity (95% CI) | | 75.0% (62.6–84.9%) | 100% (94.4–100.0%) | 100% (94.4–100.0%) |
| Specificity (95% CI) | | 100% (94.4–100.0%) | 100% (96.9–100.0%) | 100% (94.4–100.0%) |
| *p-value* | | | $p = 0.0036^a$ | $p < 0.0001^a$ |
| | | | $p = 1.000^b$ | |

Note: CI, confidence interval; 'a', statistically significant difference for the comparison between crude antigens and TPx-1 or TER ATPase; 'b', statistically significant difference for the comparison between TPx-1 and TER ATPase.

**Table 4. The performance of TPx-1 and TER ATPase in diagnosis of human echinococcosis.**

| | No. of samples | No. of positive samples | | |
|---|---|---|---|---|
| | | Crude antigens | TPx-1 | TER ATPase |
| Echinococcosis patients | | | | |
| Healthy donor | 92 | 8 | 6 | 4 |
| Patients | 90 | 58 | 60 | 64 |
| AUC (95% CI) | | 0.893 (0.845–0.941) | 0.936 (0.898–0.963) | 0.976 (0.960–0.993) |
| Sensitivity (95% CI) | | 64.64% (53.65–74.26%) | 66.67% (56.82–76.80%) | 71.11% (61.39–80.64%) |
| Specificity (95% CI) | | 98.91% (94.09–99.97%) | 94.44% (87.51–98.17%) | 95.65% (89.01–98.78%) |
| *p-value* | | | $p = 0.5053^a$ | $p = 0.002^a$ |
| | | | $p = 0.002^b$ | |

| | No. of samples | No. of positive samples | | |
|---|---|---|---|---|
| | | Em18 | TPx-1 | TER ATPase |
| Cured echinococcosis patients | | | | |
| Cured patients | 59 | 5 | 10 | 11 |
| AUC (95% CI) | | 0.976 (0.942–0.993) | 0.881 (0.82–0.93) | 0.952 (0.90–0.98) |
| Sensitivity (95% CI) | | 92.19% (82.98–96.62%) | 83.05% (71.03–91.56%) | 81.11% (71.49–88.59%) |
| Specificity (95% CI) | | 94.62% (86.08–98.61%) | 77.78% (67.79–85.87%) | 93.22% (83.54–98.12%) |
| *p-value* | | | $p < 0.001^a$ | $p < 0.001^a$ |
| | | | $p = 1.000^b$ | |

Note: CI, confidence interval; 'a', statistically significant difference for the comparison between crude antigens or Em18 and TPx-1 or TER ATPase; 'b', statistically significant difference for the comparsion between TPx-1 and TER ATPase.

from the parasite to the host, or from the parasite to environment where it lives [8,27]. In the present study, we performed proteomic analysis of serum-derived EVs from *E. multilocularis*-infected mice and found the constant existence of two parasite-origin proteins in serum EVs. Our results identified both TER ATPase and TPx-1 as promising candidates for the diagnosis of echinococcosis in humans and animals.

Consistent with the previous reports [28,29], the enriched EVs were spherical in shape, with a diameter ranging from 50 to 200 nm. The enriched EVs was further confirmed by detection of EV marker CD63. These data indicated that we isolated highly pure serum EVs. The proteomics data showed that the abundance of 18 proteins in the EVs was altered significantly in response to *E. multilocularis* infection. Extensive proteomics has verified the presence of TGF-β1 in EVs [30,31], which was recently localized on the surface of most cell-derived EVs [32]. TGF-β is a multifunctional cytokine that is involved in a variety of infectious diseases [33,34]. TGF-β is elevated during parasite infection, and contributes to survival by suppressing immune responses [35]. It would be very interesting to pinpoint the role of TGF-β1 during *E. multilocularis* infection.

There is growing interest in identifying important parasite-derived molecules for use as diagnostic candidates [36]. Parasitic helminths are able to release biologically active molecules into their host tissues [37]. Therefore, parasite-origin proteins are normally found in the host's blood plasma and sera during infection [38,39]. The significance of exosome-derived proteins as biomarkers for infectious diseases has been gradually recognized [40,41]. In the present study, ten *E. multilocularis*-origin proteins were identified in serum-derived EVs from infected mice. Among them, some proteins, including TPx-1, FBPA and GAPDH have already been identified in EVs from protoscolex culture supernatant (PCS-EVs) [15]. Moreover, some parasite derived-proteins in EVs from sera/plasma of CE patients or plasma of *E. granulosus*-infected mice were also simultaneously present in PCS-EVs or HCF-EVs [39,42,43], suggesting that EVs released by *E. granulosus* enter into the the circulatory system. In the present study, TPx-1 and TER ATPase were verified to be present in serum EVs during *E. multilocularis* infection. A previous study showed that TPx-1 could be a promising circulating antigen for the diagnosis of helminthic diseases, such as schistosomiasis [44]. TER ATPase, a member of AAA-ATPase family, is involved in endoplasmic reticulum membrane homeostasis and ubiquitination [45], and the prognosis of cancer [46]. Consistent with the findings of the present study, TER ATPase has also been reported in EVs released from various cancer cells [47] and its downregulation greatly reduces exosome secretion, suggesting targeting of TER ATPase as a new anti-tumor therapy approach [48]. Moreover, TER ATPase can serve as a biomarker for the development of pathology at the early clinical and preclinical stages of Parkinson's disease in humans [49]. However, the role and diagnostic values of the EV TPx-1 and TER ATPase of *E. multilocularis* have not been investigated.

The finding of their continuous presence in serum EVs prompted us to evaluate TPx-1 and TER ATPase as candidates for early serologic diagnosis of echinococcosis. Both TPx-1 and TER ATPase were specifically recognized by sera of *E. multilocularis*-infected mice, suggesting that they are immunogenic. Compared with crude antigens, TPx-1 and TER ATPase performed better in early diagnosis, being able to detect experimental murine AE as early as 10 days post infection. Moreover, the sensitivity and specificity of TER ATPase were also superior to diagnose human echinococcosis compared with those of TPx-1 and crude antigens. Given the unavailability of serum samples, it is still unclear whether TER ATPase is suitable for early diagnosis of human echinococcosis. The Em18 antigen is known to be a valid marker for follow-up [50]. Our results showed that, for cured patients, both TPx-1 and TER ATPase had a higher positive rate than the recombinant Em18 protein. Consistent with previous hypotheses [44], this finding could be explained by fact that the release of both EV antigens being closely

related to the physical conditions of parasites. Therefore, it would be very interesting to investigate the relationship between the levels of specific antibodies against TPx-1 and TER ATPase and prognosis. Taken together, these results suggest that both TPx-1 and TER ATPase are promising diagnostic markers for evaluation of the prognosis of cured patients.

In summary, we unveiled the dynamic changes in protein cargoes in serum EVs in response to *E. multilocularis* infection and identified three parasite-origin proteins that were continuously present in the serum EVs. We further verified TPx-1 and TER ATPase as promising early/prognostic diagnostic markers of echinococcosis.

## Supporting information

**S1 Fig. Preparation of polyclonal antibodies against recombinant TPx-1 or TER ATPase.**
(A) and (B) SDS-PAGE analysis of purified recombinant TPx-1 or TER ATPase (truncated).
(C) Western blotting analysis of native TPx-1 using anti-TPx-1 polyclonal antibodies. (D)
Western blotting analysis of native TER ATPase using anti-TER ATPase polyclonal antibodies.
(TIF)

**S1 Table. Primers used for the generation of the recombinant proteins.**
(DOCX)

**S2 Table. Summary of Proteins identified of serum-derived exosomes.**
(DOCX)

**S3 Table. The performance of TPx-1 and TER ATPase in diagnosis of animal echinococcosis.**
(DOCX)

## Author Contributions

**Conceptualization:** Xiaola Guo, Shuai Wang, Yadong Zheng.

**Formal analysis:** Xiaola Guo, Junmei Zhang.

**Funding acquisition:** Xiaola Guo, Yadong Zheng.

**Investigation:** Xiaola Guo, Rui Li.

**Methodology:** Yong'e Zhang.

**Resources:** Zhengrong Wang, Qingming Kong, Yujuan Shen, Lingqiang Zhang, Haining Fan.

**Software:** Xiaola Guo, Junmei Zhang.

**Supervision:** Xiaola Guo, Yadong Zheng.

**Validation:** Yadong Zheng.

**Visualization:** Xiaola Guo.

**Writing – original draft:** Xiaola Guo, Shuai Wang, Yadong Zheng.

**Writing – review & editing:** Xiaola Guo, Shuai Wang, William C. Cho, Xianghong Ju, Jianping Cao, Yadong Zheng.

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
