## [Decision Letter · Decision Letter 0]

5 Jul 2022

Dear Dr Guo,

Thank you very much for submitting your manuscript "Proteomic profiling of serum extracellular vesicles identifies diagnostic markers for echinococcosis" for consideration at PLOS Neglected Tropical Diseases. As with all papers reviewed by the journal, your manuscript was reviewed by members of the editorial board and by several independent reviewers. In light of the reviews (below this email), we would like to invite the resubmission of a significantly-revised version that takes into account the reviewers' comments. 

The three Reviewers have found that the article is of potential value. At the same time, they have raised a number of major questions and concerns, which must be addressed by the authors very carefully for the article to be acceptable. These questions and concerns include (but are not limited to): (i) whether or not the candidate diagnostic proteins were detected in EVs from non-infected mice; (ii) the need to compare the results with published proteomic data for EVs in Echinococcus infection and in supernatants of E. multilocularis parasite culture; (iii) the need for explicitation of the immunogold labeling methods use (and the need to show appropriate controls in these results); (iv) the reason why the 14-3-3 protein was not followed up as a possible diagnostic marker; (v) the parasite or host specificity of the antibody used for detection of the 14-3-3- protein; (vi) the need to acknowledge the limitations of the EV enrichment/purification method used; (vii) the need for ethical approval for the use of human samples.

We cannot make any decision about publication until we have seen the revised manuscript and your response to the reviewers' comments. Your revised manuscript is also likely to be sent to reviewers for further evaluation.

Sincerely,

Alvaro Diaz, PhD

Associate Editor

Adriano Casulli

Deputy Editor

Reviewer's Responses to Questions

**Key Review Criteria Required for Acceptance?**

**Methods**

-Are the objectives of the study clearly articulated with a clear testable hypothesis stated?

-Is the study design appropriate to address the stated objectives?

-Is the population clearly described and appropriate for the hypothesis being tested?

-Is the sample size sufficient to ensure adequate power to address the hypothesis being tested?

-Were correct statistical analysis used to support conclusions?

-Are there concerns about ethical or regulatory requirements being met?

Reviewer #1: Subtitle: Human serum samples:

At the beginning of M&M section, authors state ethical aspects regarding animal experiments but they did not specify anything regarding human samples.

¿There any public/state/country regulation regarding this point? ¿Do the patients have agreed and/or signed any kind of consent to the use of their sera? ¿Have the samples be anonymized? Please add this information in “Ethic statement” section.

Subtitle: Parasite infection

Reading the entire article I realized (or I supposed) that the authors pooled the sera of 10 mice of each group giving rise to three of EV-infected mice samples and three of EV-uninfected mice samples, for each time point. Is, in fact, the experiment designed in this way?

In the same sense, in line 105 it is stated that the sera of 10 uninfected mice were harvested together. But as the sentence only refers to uninfected mice, I did not understand the reason of pooling just those samples. It is possible that in this sentence the EV-infected mice were involuntary omitted?

Please re-write this section to clarify from the very beginning which groups are going to be compared and if the experiment is well-designed.

Subtitle: Purification and characterization of EVs

The authors describe the purification of EV. Finally the EV are resuspended either in PBS or denaturation buffer. I wonder if the authors perform any additional step to lyse the EV and to recover protein samples to be used in proteomic studies. I think that this step is mandatory. Only resuspending in any denaturating buffer can lead to membrane lyses but much better yield when a specific lysis procedure is performed.

Subtitle: Protein digestion and mass spectrometry

The authors describe reduction and alkylation reaction routinely carried in MS sample prep. Normally 10 mM DTT is used and an excess of IAA (typically 50-55 mM). The reason for IAA excess is to assure that react with all sulfhydryl group in proteins and in the reaction mixture. Remember that DTT has two sulfhydryl groups and for that reason at least the concentration of IAA normally used is at least twice. Please confirm if the concentration of IAA expressed in the text is correct.

In the digestion step:

1- Specify the brand and quality of trypsin used and the relation enzyme to proteins used.

2- The authors add a subsequent additional step of digestion. Trypsin used for proteomic studies is very robust and efficient enzyme. This is not the common procedure. Do the authors have any particular evidence of incomplete protein digestion that explain this extra step?

LC-MS/MS analysis:

For nanoHPLC specify the following parameters:

• How is the relation of ACN/FA in mobile phase A and B.

• Which column was used in nano-HPLC system (diameter, length, size of pore, C18).

• Gradient used in nano-LC system.

For MS analysis specify the following parameter.

• MS voltage setting, capillary temperature.

• Full mass scan parameters: ranges of m/z acquisition, resolution.

• Fragmentation parameters: number TopN fragmented ions in each duty cycle, collision energy, and resolution.

Subtitle: Protein identification

Add the following information:

• The dates when de protein databases were downloaded.

• Fixed and variable modification of peptides.

Subtitle: Preparation of recombinant proteins

Does any tag was added to protein sequence in the cloning process to assure protein purification? If yes, please clarify this point.

Subtitle: statistical analysis

It is clear in other section of M&M and later in the paper, but it should be more clarifying if you detail which are the “three groups” mentioned in this section (30 d, 60 d, and 90 d infected and uninfected mice?).

Reviewer #2: Please state the source and concentration of trypsin used for the digests.

Line 226 the term “exosomal” here is incorrect. We do not know what type of vesicles have been isolated here -in all likelihood there will be a mix of vesicle types pelleted by ultracentrifugation. I think “EV” should be used instead (here and elsewhere in the manuscript).

Figure 2 – immunogold labelling. No methods for this are described – please add.

Reviewer #3: The manuscript by Guo and colleagues describes serum-derived extracellular vesicles screened in model-organisms and patients infected with Echinococcus multilocularis, aiming in an early detection of the infection. The study design is appropriate. However, a few details are not described in sufficient details to follow all the conclusions of the manuscript.

Line 107ff. Please provide more details about the human samples. How are the parasite cysts staged? Which IHA test was performed. Which follow-up marker was used and how were the patients verified to be “cured”? For example an Em18-test should be performed, as this is currently the best marker for viable and non-proliferative infections (see the corresponding PLoS NTD article: https://doi.org/10.1371/journal.pntd.0010146, which should be included in the discussion). Furthermore, it has been also shown, that various tests (like the Em18), but also other tests are not good for the early and sub-clinical detection of the parasite (see https://doi.org/10.3390/pathogens11050518).

Line 114ff. According to the MISEV guidelines on how studies on extracellular vesicles should be described (https://doi.org/10.1080/20013078.2018.1535750), ultracentrifugation only provides an enrichment of the particles. Therefore, this should be clearly stated, that this is an “enrichment” and not a purification. As this level of purity is sufficient for the further conclusions, some hits (like TGF-b) are probably still serum-derived and not from the host EV cargo. This should definitively be discussed according to the guidelines.

Line 142ff. Please provide the genome and UniProt release number

Line 165. Provide the number of used rabbits.

Line 179. For what purpose where X-ray films used? Please describe the imaging, as ECL is usually a chemiluminescent dye.

Line 180ff. Please describe the ELISA in more details. Which sera were used? From which species in which experiment? How many samples per group? How were they collected?

Line 194ff. Please describe how the positive/negative threshold was determined? Was it by a ROC and maximal youden’s index or by S/N values?

**Results**

-Does the analysis presented match the analysis plan?

-Are the results clearly and completely presented?

-Are the figures (Tables, Images) of sufficient quality for clarity?

Reviewer #1: Subtitle: Mouse protein components in serum-derived EVs

Page 10, line 219: Since all proteomic analysis software apply statistical analysis is more correct to express “were statistically differentially abundant…” or “show a statistically difference in relative abundance…”.

Subtitle: Parasite-origin protein components in serum-derived EVs

In this section the authors describe a series of proteins that are consistently identified in EV obtained from infected mice. They do not correspond with the most abundant statistically differential proteins identified in the first result’s section. The proteins describe do not necessarily have to be the most abundant, but… they are identified as statistically differential? Please, add a comment referring to this point.

Most importantly, the author designed a mice experiment accounting considerable number of infected and uninfected mice and performed a comparative proteomic analysis. As a result, they described the presence of protein detected in E. multilocularis infected-EV mice sera in all time points studied. But they do not mention at all if those proteins are detected or not in uninfected samples. The rest of the paper centers in the validation of those proteins that are described as good proteins for diagnosis and prognosis. For the reasons mentioned above I consider really important to know if the selected proteins are detected in the uninfected samples. Please describe, as detailed as possible, this point in this result section (include this information in the main text and tables).

It should be very interesting also that the validation experiments by WB include EV derived from uninfected mice and confirm the absence of signal. Consider that uninfected samples are the best negative control to the conclusion of the present work

Subtitle: Values of TPx-a and TER ATPase in the early diagnosis of AE

The authors test the immunogenicity of those proteins using sera of infected mice. In this section it would also be absolutely interesting to study this response in uninfected mice samples as the best negative control. Experiment using uninfected mice sera should be performed, present in the main text, figures and discussion section.

Figure legends:

Fig.1: Figure captions should understandable by itself without the need to come back to main text. Specify that CD63 is an exosomal marker. Also, when “those proteins” is written, is better to name it in the legend text.

Reviewer #2: Figure 2. I assume antibody-conjugated gold probes were used as secondary antibody and that this is what the arrow is pointing to in Figure 2? The legend should be revised to include more description. What is the difference between Fig 2A and Fig 2B? Do they both show the same thing? I can only see a single probe (arrowed) which is not very convincing. Additional images showing more probes, specifically clustered on the EVs, is required. 

A mouse 14-3-3 protein was also identified in the EVs (Table S2). How do you know if the Em-14-3-3 antibody is specific for parasite-derived EVs? Some images where this antibody has been applied to serum EVs from uninfected mice should be included.

Figure 2 legend – panel D is not described.

Have the authors tried immunolocalization using their antibodies on tissue sections of E. mulitlocularis in situ within host tissues? It might help to understand how the parasite is releasing EVs and how they end up in the host blood.

Reviewer #3: Line 214. Please specify, that CD63 was an mammalian EV-marker, as there is several evidence, that the tetraspanins are not cross-reacting between helminth and other species. See e.g. Sotillo et al., 2020: https://doi.org/10.1016/j.ijpara.2020.04.010

Line 274. The crude Em antigen is known not to be a valid marker for follow-up. Rather use the state-of art diagnostics for distinguishing viable vs non-productive infections, such as the global available Em18 antigen. This must be included to draw the conclusions to define “cured” patients.

**Conclusions**

-Are the conclusions supported by the data presented?

-Are the limitations of analysis clearly described?

-Do the authors discuss how these data can be helpful to advance our understanding of the topic under study?

-Is public health relevance addressed?

Reviewer #1: Conclusions are partially supported by the data presented. Very relevant conclusion are obtained. The authors engage in a big work, using a beautiful set of samples (EV from sera of a total of 120 mice –infected and uninfected-, and sera from sick and healthy humans), but negative samples obtained from uninfected mice is not discussed and is not used as negative control in very relevant experiments. In my opinion, those samples are the best negative control and the authors do not take advantage of it. The authors should repeat the experiments using those extremely valuable samples.

Also, the proteomic detection of the protein selected as relevant for prognosis and diagnosis in negative samples must be discussed in Results and Discussion sections.

Reviewer #2: Generally OK but more could be added if the suggested proteomics comparisons were included.

Reviewer #3: The data presented by Guo and colleagues provide important results to diagnose accurately this important disease in humans. Some limitations are described. Some aspects however are missing and should be included:

It is somehow unclear, why the important EV protein “14-3-3” was not considered as a candidate, beeing a known and specific EV-marker. Is there any further data on this? Please comment on that, as this protein is stated in the abstract as one of the promising candidates, but on which no further experiments were performed.

Line 288 / 295. There is no data showing the purity of EV’s. The used technique only allow to enrich EVs. A contamination with soluble excreted/secreted proteins is probable, such as for example the abundance of TGFb. Further details about purity and working with helminth EVs are provided in Sotillo et al., 2020: https://doi.org/10.1016/j.ijpara.2020.04.010, and the MISEV2018 guidelines (https://doi.org/10.1080/20013078.2018.1535750).

Line 321. The early diagnosis was only shown in murine samples, which were infected i.p., which is a highly artificial model. Therefore, this conclusion should be put into perspective of the used system.

**Editorial and Data Presentation Modifications?**

Reviewer #1: Main page; Abstract: Echinococcus in italics.

 E. multilocularis in italics

 E. multilocularis –infected mice, without space between multilocularis and mid-dash.

Page 1, line3: Wang with capital letter.

Page 5, line 88: “proteomic” instead of “proteomics”.

Page 5, line 96: “animal” instead of “Animals”.

Page 6, line: 116: “group” instead of “groupe”.

Page 6, line: 124: “1:5” instead of “1: 5”, without space.

Page 6, line 131: is better to used IAM as an abbreviation of iodoacetamide, since IAA is frequently used also to iodoacetic acid. To avoid confusion, please use IAM.

Page 7, line 132: express the temperature as 37 °C, with an space between the number and the unit.

Page 7, line 151: add a space between “<” and “0.05”.

Page 8, line 172: add a space between “4” and “°C”, and delete the space between “1” and “:” in “1:10,000”.

Page 8, line 173: delete the space between “USA)” and “,”.

Page 9, line 183, 185, 186: Add a space between the number and “°C”.

Page 9, line 186, 188: delete the spaces between “:” and the number that follows.

Page 22, line 462: “from” instead of “fromr”.

Page 28, line 533: “E. multilocularis” in italics.

Reviewer #2: Some areas might need to be checked for grammar/spelling.

Reviewer #3: Line 70: E. multilocularis and E. granulosus sensu latu are the only causative agents, and not the most important ones. Please change this.

Line 81. It is right, that EVs are first suspected to be a garbage-disposal system, however, there is plenty of evidence, that they are specific interactors. I suggest to change the sentence to “EVs were orginally identified in reticulocytes as a supposed garbage-disposal mechanism.

Line 95. Please change the sentence to “has been approved by the ...”

Line 116. An “r” is missing in “group”

Line 130. Please write “EV proteins” (EV without “s”)

Line 154. Please delete the “o”

Line 167. The IgG are polyclonal. Therefore, not only specific antibodies are purified by Prot G Sepharose. Therefore, “specific” should be deleted.

Line 259. I suggest adding the p-value, and omit in the table 3-

Line 266. I suggest deleting “approximately”, as it was exact the stated number.

Line 267. I would suggest adding the p-value (p=0.002), and omit in the table

Line 271. It is rather a “prognostic value” as a “diagnostic value”

Line 325. Just write “is suitable for early diagnosis of human echinococcosis”.

Figure 2. Please indicate in the figure capture, what the red arrow means. The approach of immune-gold labelling is missing in the material and methods sections and should be described in more details.

**Summary and General Comments**

Reviewer #1: The paper describes the identification, by a proteomic approach, of two relevant proteins useful for early diagnosis and prognosis of echinococcisis. The data is validated by different methodologies. The experimental design is very well performed using EV derived from E. multilocularis infected and uninfected mice. They use a significant number of animals and different time points. They also use human sera from sick and healthy humans. In principle, it seems a wonderful work, but unfortunately the authors do not uses the uninfected mice proteomic data in discussion. Also they do not take advantage of these invaluable samples as the best negative control for all the validation strategies. I strongly consider that negative samples derived resulted must be discussed and also consider that those sample should be used in validation experiments as negative controls.

Reviewer #2: Given the amount of proteomic data that is available for EV from related species (E. granulosus), either purified EVs or isolated from mouse sera, cyst fluid etc (see links to examples below – there are others) the authors have missed an opportunity for comparison of parasite-derived EVs detected in host fluids.

https://pubmed.ncbi.nlm.nih.gov/35360110/

https://pubmed.ncbi.nlm.nih.gov/33708201/

https://pubmed.ncbi.nlm.nih.gov/33017416/

Also, it would be useful to compare the proteome of EVs directly recovered from E. multilocularis following in vitro culture (https://pubmed.ncbi.nlm.nih.gov/28215864/) with those isolated from serum of infected mice. A significant overlap would provide more support that parasite-derived EVs are actually present in the mouse serum described here.

Reviewer #3: The manuscript is well presented, and the topic and work are definitively interesting for the journals’s readers and suitable to be published in PLoS NTD with some modifications. An early detection of alveolar echinococcosis is missing and would improve the treatment management. The data are presented clearly, and the manuscript is well written.

Human ethic approval is missing and should be amended, including a statement, that the consent of the patients is present.

PLOS authors have the option to publish the peer review history of their article (what does this mean?). If published, this will include your full peer review and any attached files.

Reviewer #1: No

Reviewer #2: No

Reviewer #3: No
---

## [Decision Letter · Decision Letter 1]

9 Sep 2022

Dear Dr Guo,

Thank you very much for submitting your manuscript "Proteomic profiling of serum extracellular vesicles identifies diagnostic markers for echinococcosis" for consideration at PLOS Neglected Tropical Diseases. As with all papers reviewed by the journal, your manuscript was reviewed by members of the editorial board and by several independent reviewers. The reviewers appreciated the attention to an important topic. Based on the reviews, we are likely to accept this manuscript for publication, providing that you modify the manuscript according to the review recommendations. 

The three Reviewers consider that the manuscript has much improved. The authors now have to make the minor corrections indicated by Reviewers 1 and 3, as well as make sure that their primary data are fully available for other researchers.

Sincerely,

Alvaro Diaz, PhD

Academic Editor

Adriano Casulli

Section Editor

The three Reviewers consider that the manuscript has much improved. The authors now have to make the minor corrections indicated by Reviewers 1 and 3, as well as make sure that their primary data are fully available for other researchers.

Reviewer's Responses to Questions

**Key Review Criteria Required for Acceptance?**

**Methods**

-Are the objectives of the study clearly articulated with a clear testable hypothesis stated?

-Is the study design appropriate to address the stated objectives?

-Is the population clearly described and appropriate for the hypothesis being tested?

-Is the sample size sufficient to ensure adequate power to address the hypothesis being tested?

-Were correct statistical analysis used to support conclusions?

-Are there concerns about ethical or regulatory requirements being met?

Reviewer #1: I would like to thank the authors for addressing all of my concerns in the M&M section. I found the experimental design and methodology much more clear. 

However, I still have some minor comments.

Line 122: As both set of mice (infected and unifected) were used for further experiments, I find better to delete "the infected mice" and write only "Mice were euthanized..."

Line 150: In the previous version of the manuscript it was explained that EVs-containing pellets were resuspended either in PBS or denaturing solution. In the present text the denaturing solution was omitted. I wonder why since it is a better solution to extract proteins from vescicles. And I still do not know if a protein extraction procedure was employed for proteomic analysis. Please, clarify this point.

In the previous version of the manuscript it was explained that EVs-containing pellets were resuspended either in PBS or denaturing solution. In the present text the denaturing solution was omitted. I wonder why since it is a better solution to extract proteins from vescicles. And I still do not know if a protein extraction procedure was employed for proteomic analysis. Please, clarify this point.

Line 171: If protein extraction procedure was carried out it could also be described here.

Line 171: I think that it is tidier to write the sentence "Protein concentration was measured...", before reduction and alkylation procedure description.

Line 173: It is described that 100 ug of proteins were digested (this is a quite big quantity of proteins for protomic analysis). Please specify in line 176, how much protein was injected into the mass spectrometer.

Line 179: Since the gradient described here is not only the increase phase B gradient for protein separation, but also the gradient to let the column in a standby state (a phase B decreasing gradient), please delete the word "increase".

Line 258: I asked for the specification of the "three groups" here. In the answer to the reviewers it is stated that it was done. But I didn't find it in this section. Please add it.

Data availability: the deposited data is not fully available.

Reviewer #2: (No Response)

Reviewer #3: Line 117. First sentence still unclear, what is the intension. Maybe better to write (if my suggestion is correct) "Mice were injected at two different timepoints intraperitoneally with E. multilocularis protoscoleces as described previously"

Line 124ff. If sera of 10 mice of each group were pooled, this gives duplicates (and not triplicates) for each timepoint. Please clarify.

Line 191ff. Please profide the identifier of the mice database and which genome was used for E. multilocularis. I guess it is PRJEB122? I suggest following sentence "mouse protein database (88,027 sequences) obtained from Uniprot (Release 2022/02), and E. multilocularis genome and protein databases PRJEB122 (10,971 sequences; retrieved from WormBase ParaSite)".

Line 220. Please describe, what kind of tissue sections. Which tissue, origin, ...

**Results**

-Does the analysis presented match the analysis plan?

-Are the results clearly and completely presented?

-Are the figures (Tables, Images) of sufficient quality for clarity?

Reviewer #1: Line 287: Please, also explain the behavior of AMY1 in the WB analysis.

Line 297 or 306: Explain the reason(s) why the protein 14-3-3 was excluded from further analyses.

Line 317: Is is possible that "fig. 2" here is in fact "fig. 3"?

Regarding my request to add a statistics for the proteins uniquely detected in infected samples, I checked MQ software and I must recognized that I couldn't find the option to do that. I just want to comment that the software I use to proteomic analysis (doi: 10.1038/s41596-022-00690-x) has this option. I think that it is a good option. I know that biologically, is not possible to found parasite proteins in unifected samples. But it is technically possible. First of all, you can find false positive (this is the reason why we apply a FDR rate value for the acceptance of proteins, and the authors did it). But also, sometimes it is recommended to mix control and samples of interest, because in long runs columns show drift in its performance. In this kind of run set up, this bias is considered. However, some abundant proteins of one group (either using appropiate blanks) can be detected in the other set of samples, usually with in one replicate or with vert low signal or spectral counts. The software I describe take this issue into account, and can statistically determine that a protein is present just in one set of sample.

Reviewer #2: (No Response)

Reviewer #3: No further comments

**Conclusions**

-Are the conclusions supported by the data presented?

-Are the limitations of analysis clearly described?

-Do the authors discuss how these data can be helpful to advance our understanding of the topic under study?

-Is public health relevance addressed?

Reviewer #1: Considering changes made along the manuscript, I find this section very well written. Every aspect of the results is discussed and conclusions are very well understood.

Reviewer #2: (No Response)

Reviewer #3: No further comments

**Editorial and Data Presentation Modifications?**

Reviewer #1: Line 50: delete the space between ") ,".

Line 77: "identify of diagnostic", delete "of".

Line 140: Add an space between ELISA and [.

Line 160: "Briefly" insted of "Brefily".

Lines 184-185 and everywhere is written: m/z in italics.

Line 185: add a space between the number and "m/z".

Line 222: To be consistent in the way of PBS-T is written latter in the text, write here "TBS-T".

Line 223: Add a space between "study)" and "After". The letter A is missing in "Alexa".

Line 228: "Protein samples" with lower case.

Line 269: Add a space between "1A" and "and".

Line 273: Detele the "n" from "an".

Line 273: The word enriched is writen twice very closely in the same sentence. I suggest to replace the first "enriched" whit "more abundant".

Line 351: Add a space between "EVs" and "[".

Line 351: I believe that it is "most" instead of "mast".

Line 388: Add a space between "follow-up" and "[50].

Line 390: Delete one of the two "the".

Line 408: Delete one of the two dots written together.

In the text and legends the protein "G-Patch domain containing 8" is abbreviated as GPATCH8, but in figures as "GPTC8". Please unify the nomenclature.

Reviewer #2: (No Response)

Reviewer #3: Author summary:

Line 77ff. This sentence is a bit complicated. Maybe change to "This study signify the role of EVs for the identification of iagnostic candidates by the discovery of two identified proteins, TER ATPas and TPX-1. First results indicate their diagnostic and prognostic values in experimental murine and human echinococcosis."

Materials and methods:

Line 121. A "comma" is missing before ", and the other group [...]"

Line 129. Please write "blood samples" instead of "bloods"

Line 130. A "comma" is missing before the "and": "post infection, and [...]"

Line 133. "until further use" instead of "for late use"

Line 136. "clinical" instead of "clinica"

Line 144. "until further use" instead of "for late use"

Line 160. "Briefly, [...]"

Results:

Line 273. Here, the word "enriched" is not needed.

Discussion:

Line 348. "EV marker"

Line 357. remove "including blood". It is not really needed.

PLOS authors have the option to publish the peer review history of their article (what does this mean?). If published, this will include your full peer review and any attached files.

Reviewer #1: No

Reviewer #2: No

Reviewer #3: No

**Summary and General Comments**

Reviewer #1: With the changes made the manuscript has improved a lot. All sections are much better understood, leading to a better understanding of the conclusions.

Reviewer #3: This is a second review of the manuscript submitted by Guo and colleagues. Early diagnosis of echinococcosis is missing. Therefore, the demonstrated work will provide valuable information which will guide for future point of care diagnostics to control this devastating infection disease. The revised manuscript include now more information to follow the methods and conclusions.

Figure Files:

Data Requirements:

Reproducibility:

References

---

## [Editor Report · Decision Letter 2]

14 Sep 2022

Dear Dr Guo,

We are pleased to inform you that your manuscript 'Proteomic profiling of serum extracellular vesicles identifies diagnostic markers for echinococcosis' has been provisionally accepted for publication in PLOS Neglected Tropical Diseases.

Best regards,

Alvaro Diaz, PhD

Academic Editor

Adriano Casulli

Section Editor

---

## [Editor Report · Acceptance letter]

3 Oct 2022

Dear Dr Guo,

We are delighted to inform you that your manuscript, "Proteomic profiling of serum extracellular vesicles identifies diagnostic markers for echinococcosis," has been formally accepted for publication in PLOS Neglected Tropical Diseases.

Best regards,

Shaden Kamhawi

co-Editor-in-Chief

Paul Brindley

co-Editor-in-Chief
